# Mitochondrial Transfer from Human Platelets to Rat Dental Pulp-Derived Fibroblasts in the 2D In Vitro System: Additional Implication in PRP Therapy

**DOI:** 10.3390/ijms26125504

**Published:** 2025-06-08

**Authors:** Koji Nishiyama, Tomoni Kasahara, Hideo Kawabata, Tetsuhiro Tsujino, Yutaka Kitamura, Taisuke Watanabe, Masayuki Nakamura, Tomoharu Mochizuki, Takashi Ushiki, Tomoyuki Kawase

**Affiliations:** 1Tokyo Plastic Dental Society, 2-26-2 Oji, Kita-ku, Tokyo 114-0002, Japan; 1946kohjinishiyama6491@gmail.com (K.N.); tmntmn600@gmail.com (T.K.); hidei@b-star.jp (H.K.); tetsudds@gmail.com (T.T.); shinshu-osic@mbn.nifty.com (Y.K.); watatai@mui.biglobe.ne.jp (T.W.); maoh4618@me.com (M.N.); 2Department of Orthopaedic Surgery, Graduate School of Medical and Dental Sciences, Niigata University, Niigata 951-8510, Japan; tommochi121710@gmail.com; 3Division of Hematology and Oncology, Graduate School of Health Sciences, Niigata University, Niigata 951-9518, Japan; tushiki@med.niigata-u.ac.jp; 4Department of Transfusion Medicine, Cell Therapy and Regenerative Medicine, Niigata University Medical and Dental Hospital, Niigata 951-8520, Japan; 5Department of Hematology, Endocrinology and Metabolism, Faculty of Medicine, Niigata University, Niigata 951-8510, Japan; 6Division of Oral Bioengineering, Graduate School of Medical and Dental Sciences, Niigata University, Niigata 951-8514, Japan

**Keywords:** platelets, mitochondria, transfer, fibroblasts, in vitro, platelet-rich plasma

## Abstract

Platelet mitochondria have recently been increasingly considered “co-principal” along with platelet growth factors to facilitate tissue regeneration in platelet-rich plasma therapy cooperatively. To develop a convenient method to test this potential, we examined mitochondrial transfer using a simple two-dimensional culture system. Living human platelets were prepared from PRP obtained from 12 non-smoking healthy male adults (age: 28–63 years) and suspended in medium. Platelet lysates were prepared from sonicated platelet suspensions in PBS. After treatment with ultraviolet-C irradiation, a mitochondrial respiration inhibitor, or a synchronized culture reagent, rat dental pulp-derived fibroblasts (RPC-C2A) were co-cultured with platelets or platelet lysates for 24 h. Mitochondrial transfer was evaluated by visualization using a fluorescent dye for mitochondria or an antibody against human mitochondria. Ultraviolet-C-irradiated cells substantially lost their viability, and treatment with living platelets, but not platelet lysates, significantly rescued the damaged fibroblasts. Fibroblast mitochondria appeared to increase after co-culture with resting platelets. Although more microparticles existed around the platelets on the fibroblast surface, the activated platelets did not show significant increases in any parameters of mitochondrial transfer. This simple co-culture system demonstrated mitochondrial transfer between xenogeneic cells, and this phenomenon should be considered as an additional implication in PRP therapy.

## 1. Introduction

Based on the concept that platelet-rich plasma (PRP) accelerates wound healing and tissue regeneration by activating tissue and circulating stem cells through concentrated growth factor cocktails, PRP therapy has spread rapidly and is widely used in many other medical fields [1]. This growth factor-centered concept was considered absolute during the historical early phase and was accepted without great suspicion. However, as disappointing clinical outcomes have often been reported, various questions have arisen [2]. First, owing to the differences in individual patients’ blood samples, preparation protocols, and operators’ skills, similar PRP quality cannot be guaranteed, even if they are prepared from the same samples. Second, PRP may not be effective alone, regardless of local and systemic physical conditions. Instead, PRP works as an “adjuvant” to augment the effects of the preceding surgical operation, medication, or the patient’s regenerative capability [3,4], Third, the clinical outcomes of PRP therapy may be significantly modulated by the levels of anti-inflammatory factors in the individual PRP preparations [5,6]. In response to the first question, the necessity of standardizing the preparation protocol was proposed and investigated to determine the optimal procedure [7,8,9,10,11,12,13]. Similar actions are required in the clinical treatment protocol [14,15,16]. Other questions have been investigated to overcome the considerable variations in clinical outcomes and to improve PRP therapy.

PRP contains numerous factors that may influence the clinical outcomes of PRP therapy. Since PRP therapy was first reported [17], many factors, including age, sex, and blood type, have been identified as possible influential factors. To our knowledge, age and sex have been reported to influence growth factor levels in PRP [18,19,20,21]. However, their correlation with clinical outcomes remains unclear. Recently, a remarkable advance in proteomic analysis has comprehensively demonstrated the proteins that regulate the efficacy of PRP [22,23,24,25,26]. As described above, anti-inflammatory factors and plasma factors, as well as pro-inflammatory cytokines, are promising candidates [27,28,29]. However, neither factor seems sufficient to explain the unexpected outcome or existence of “non-responders” to PRP therapy.

To break this deadlock, a new concept has recently been introduced, complementing the classic concept, which is solely dependent on growth factors. This concept involves mitochondrial transfer [30,31,32]. It is well known that mitochondria are intracellular organelles involved in energy (ATP) generation to maintain cellular functions in eukaryotic cells [33]. Mitochondrial mass and quantity are generally thought to be related to the type and level of cellular functions. Muscle cells contain many mitochondria that meet high energy demand [34]. In platelets, due to their small size, only 5–8 mitochondria are usually included and are thought to contribute to activation [35]. In contrast, the major power unit is glycolysis, which maintains the basic functions in the resting state in platelets [36]. Therefore, even though many platelets gather to form clots at injured sites, it is quantitatively challenging to speculate that a significant number of mitochondria can be efficiently incorporated into surrounding cells. Recent studies have demonstrated this phenomenon and its possible involvement in wound healing and tissue regeneration [30,31,32].

Independently of this recent trend, we focused on platelet bioenergetics as possible promising markers of tissue regeneration capability and investigated energy metabolism in professional and college athletes [37,38,39]. Compared with non-athletic adults, the athletes showed a relatively high involvement of oxidative phosphorylation in energy generation and maintained lower levels of platelet ATP, regardless of their performance. If mitochondrial transfer occurs at the site of PRP injection at biologically significant levels, this concept could be recognized as a “paradigm shift” and may modify the protocols for PRP preparation and clinical application. This is because the current widely accepted concept recommends the activation of PRP before injection to completely release the growth factors stored in platelets [7,40,41]. Fully and irreversibly activated platelets release growth factors and intracellular organelles, including mitochondria. Due to limited diffusion and rapid degradation in the plasma, such a release of mitochondria, regardless of whether it is encapsulated or naked, is not thought to be efficiently incorporated into surrounding cells. Thus, it could be speculated that living platelets enable more efficient transfer of their mitochondria in this situation and that prior activation of PRP may attenuate its efficacy in specific applications in which many surrounding cells are severely damaged.

To address this question, we established a simple two-dimensional co-culture system for easy evaluation. We also chose rat dental pulp-derived fibroblasts as partners of human platelets to expand the current knowledge about mitochondrial transfer.

## 2. Results

The effects of UVC on RPC-C2A cell survival and the rescuing effects of living platelets on UVC-irradiated cells are shown in Figure 1. These three groups were significantly different from each other. UVC irradiation markedly reduced cell survival in the 5% FBS-containing medium, whereas the addition of living platelets modestly but significantly rescued this reduction in cell survival. Platelet lysates prepared by sonication and filtration also significantly rescued this reduction; however, the degree of this effect was smaller than that of living platelets. These data served as the starting point of this study, with the question of whether platelets included in PRP may positively influence tissue regeneration beyond their growth factors.

The time-course changes in platelet counts in co-culture with RPC-C2A cells are shown in Figure 2. The cells were pretreated with 50 pg/mL demecolcine to stop cell division and block an unexpected reduction in mitochondrial numbers during cell division. The changes in platelet counts were recorded using a time-lapse video system from one hour after the addition of platelets to 25 h, for a total of 24 h (Appendix A). Regarding resting platelets, platelet decreases were recognized approximately 6 h after starting the time-lapse recording (7 h after beginning co-culture). Substantial platelet decreases were observed at 24 h (Figure 2A). Similarly, the platelets activated by 3 μM ADP decreased over time. No apparent differences were observed between these two groups (Figure 2B). Regarding the choice of platelet activators, we preliminarily confirmed that ADP at this concentration reproducibly increased CD62P expression (Appendix A) and induced significant platelet aggregation (Appendix A), regardless of donor. It should also be noted that the relatively large, round-shaped RPC-C2A cells found in both cultures were due to the treatment with a microtubule depolymerizing agent, demecolcine [42]. In the absence of this agent, such relatively large round-shaped cells were not found in the absence or presence of living platelets (Figure 2C,D). The relatively small, irregularly round cells in these images were recognized as typical dividing cells without growth arrest. In addition, tiny dots were observed in time-lapse recordings (Figure 2A,B) that could almost be identified as platelets (Figure 2E).

Photomicrographs of platelets and their microparticles on the surfaces of demecolcine-pretreated RPC-C2A cells, obtained by scanning electron microscopy, are shown in Figure 3. Platelet adhesion was observed in several (10%), but not all, cells when the platelets were in the resting state (Figure 3C,D). In addition, the number of resting platelets attached to the cell surface was lower than that of the activated platelets. The activated platelets released many microparticles on the cell surfaces (Figure 3E,F). In the control without platelets, no similar small or medium-sized particles were observed on the cell surfaces (Figure 3A).

The effects of living platelets (PLTs) on the number of intact mitochondria indicated by the membrane potential staining in the RPC-C2A cells are shown in Figure 4. To reduce the original mitochondrial membrane potential, the cells were pretreated with 0.1 mM FCCP. As expected, FCCP reduced the mitochondrial membrane potential (Em) in Experiment 1 (Figure 4B vs. Figure 4A). The intact mitochondria count substantially increased when living platelets were added to these cells (Figure 4C vs. Figure 4B). In the independent experiment using demecolcine (Experiment 2), adding living platelets substantially increased the intact mitochondria staining (Figure 4F,G vs. Figure 4E). The activated platelets did not show increased mitochondrial staining (Figure 4G vs. Figure 4F). The quantification data for the fluorescence signals obtained from the image analysis are shown in Figure 4H. In Experiment 1, significant differences were observed between the control and FCCP groups, as well as between the FCCP and FCCP + platelet groups. In Experiment 2, the DMC + PLTs and DMC + PLTs + ADP groups differed significantly from that of the control group. These two groups showed higher total brightness than the DMC group; however, no significant differences were observed between DMC alone and the two groups.

The effects of the living platelets (PLTs) on the distribution of human mitochondria in the RPC-C2A cells are shown in Figure 5. As theoretically anticipated, the rat cells did not display staining for human mitochondria (Figure 5A,C). However, the addition of living human platelets in the resting state increased the distribution of immunoreactive particles, that is, human mitochondria, in the cells. Immunoreactive particles were observed in the case of activated platelets (Figure 5D vs. Figure 5C); however, this distribution was much sparser than that in resting platelets (Figure 5D vs. Figure 5B).

Visualization of platelet mitochondrial transfer into RPC-C2A cells using MitoTracker reagents is shown in Figure 6. Essentially, as observed in the mitochondrial membrane potential assay and immunofluorescence staining, platelet mitochondria, stained red, were observed among green-stained mitochondria in the control intact fibroblasts (Figure 6A). UVC irradiation decreased the fibroblast mitochondrial density; however, mitochondrial transfer was not substantially affected in the UVC-damaged fibroblasts under these experimental conditions (Figure 6B).

The “three-dimensional” distribution of the transferred mitochondria in the RPC-C2A cells is shown in Figure 7. As illustrated in the cross-section of an immobilized fibroblast in the left upper panel, most of the transferred mitochondria were anticipated to exist in the cytoplasm, and a few mitochondria existed above and below the nucleus. In the upper (rows 2 and 3) and lower stages (rows 5 and 6) of the Z-stack images, a few Mi-toTracker-stained particles overlapped with the DAPI-stained nucleus region. However, in the middle stage (rows 3 and 4), the MitoTracker-stained particles did not overlap and gathered noticeably around the nucleus (i.e., the cytoplasm). This finding did not exclude the possible distribution of mitochondria on the fibroblast surface; however, since MitoTracker-stained particles are more brilliant when caged in platelets, this possibility is less likely.

The effects of the living platelets (PLTs) on the intact RPC-C2A cell counts and cellular ATP levels are shown in Figure 8. In the presence of inhibitors such as FCCP and demecolcine, the addition of living platelets did not significantly increase the intact cell counts within 24 h. However, cellular ATP levels increased significantly in the synchronized cultures treated with resting platelets (Figure 8B). Despite rough comparisons, no substantial differences were found in cell counts or ATP levels between the resting (DMC + PLTs) and activated platelets (ADP + DMC + PLTs) (Figure 8B vs. Figure 8C).

## 3. Discussion

Mitochondria are indispensable intracellular organelles in eukaryotic cells and regulate cellular functions through ATP generation. However, because of the increased oxygen required for the mitochondrial electron chain and oxidative phosphorylation, mitochondria produce reactive oxygen species (ROS), thereby damaging the mitochondria and other cell components [43]. Therefore, mitochondrial activity cannot be achieved without endogenous antioxidant enzymes.

Mitochondria also play an essential role in platelet energy metabolism. However, platelets do not require high-power units to regulate cell division. In the resting state, platelet energy is supplied mainly by oxidative phosphorylation, whereas upon activation, glycolysis demonstrates its ability to drive aggregation, exocytosis, morphological changes, and motility to stop bleeding. Therefore, the major power unit is glycolysis in platelets [36,44,45], while the mitochondrial quantity is limited in platelets and seemingly insufficient as a source of mitochondria to transfer to other cells. However, it is speculated that platelet aggregation could overcome the inferiority of the number of mitochondria per platelet by the scale effects of total aggregated platelets in vivo.

The main finding of this study was that platelet mitochondria were transferred to fibroblasts, as assessed by microscopic examination of mitochondrial staining, in a simple 2D culture system using a limited number of platelets. In addition, this transfer occurred between heterogeneous cell groups, from human platelets to rat fibroblasts. Based on the mitochondrial membrane potential and fibroblast ATP levels, a certain number of transferred mitochondria are intact and active for energy generation. In support of this scenario, in the survival assay using UVC-irradiated fibroblasts, the addition of living platelets significantly rescued the number of damaged fibroblasts, probably by replacing damaged mitochondria with intact ones. Since the rescuing effects were less apparent than expected, further investigation is needed to improve this simple co-culture system, for example, by optimizing the induction of mitochondrial damage or functional exclusion and the protocol of platelet application.

To support our interpretation based on indirect evidence, we further attempted to show the distribution of the transferred mitochondria in fibroblasts by maximizing the potential of a conventional fluorescence microscope. Most of the transferred mitochondria appeared in the cytoplasm, although relatively few mitochondria caged in platelets may have existed on the fibroblast surface.

We examined activated platelets to identify the mechanisms underlying this phenomenon, hypothesizing that microvesicles released from activated platelets may efficiently transfer mitochondria. However, mitochondria were not significantly increased in the fibroblasts treated with ADP-activated platelets compared with resting platelets. Among the four proposed mechanisms [46], “microvesicle” and “extrusion and internalization” may be excluded from the possible central mechanisms of mitochondrial transfer under these experimental conditions. Although no typical images of “tunneling nanotubes (TNT)” or “dendrites” were observed in the SEM images, contact between these two cell types seems to be a prerequisite for this transfer. Further investigations are needed to identify the mechanisms underlying this phenomenon.

### 3.1. Proposal to Clinical Protocol

The major factors that significantly influence the clinical outcomes of PRP therapy are the quality of the PRP and the patient’s physical condition. In the case of autologous PRP, as long as PRP is prepared according to a standardized protocol, PRP quality can be determined based on platelet and leukocyte counts, growth factor levels, and the choice of anticoagulants. The only point to be considered is the activation process before PRP injection [41]. If severely damaged cells are limited at the injury site, and tissue and circulating stem cells can be supplied sufficiently, the activation process is not an issue. Even if mitochondrial transfer does not occur, surrounding cells would actively regenerate injured tissue in response to PRP-supplied growth factors.

However, if the injury is severe or stem cells are not sufficiently supplied, for example, in older patients, it would be necessary to revitalize the damaged surrounding cells. In this case, mitochondrial transfer is the most potent solution to overcome this difficulty. In line with this concept, platelet mitochondria and growth factors are inseparable for successful PRP therapy. Thus, although this has not yet been clarified, we should be more careful when preparing PRP. Without consideration of the patient’s spontaneous regenerative capability, manual-based activation procedures for PRP therapy could sometimes attenuate the expected clinical outcomes of PRP therapy in patients in whom intact stem cells are not constantly supplied.

In contrast to the growth factor levels provided by PRP, a method to evaluate its potential to transfer functional mitochondria to injured cells has not yet been established. This study provides a prototype of this testing method. Further improvement is needed; however, this challenge will lead to increased predictability in PRP therapy.

### 3.2. Limitations

The injected PRP forms a clot at the injury site, where fibrin networks entrap the aggregated platelets. Therefore, many more platelets are thought to be in three-dimensional contact with the surrounding cells. Theoretically, in the 2D culture system used in this study, only one side of the fibroblast can be attached to platelets and their released microparticles, a phenomenon that should limit the incorporation of platelet mitochondria, regardless of the transfer manner.

This study employed UVC irradiation to create a model of damaged cells. It is widely accepted that ROS production induced by UVC and other factors is critical when the transition from human health to disease occurs. In addition to ROS, many other known and unknown factors can damage cells, including burns, bruises, and mechanical abrasions. However, it is challenging to control cell damage in vitro in the case of the latter factors. In a previous study [47], we compared X-ray, UVC, and hydrogen peroxide cell damage. Considering the cost, convenience, and reproducibility, we conclude that UVC is the most convenient and less expensive method. Because damaged cells are thought to be less capable of incorporating external mitochondria, depending on the levels of damage, it would be helpful to control the level of cell damage in the damaged cell model to better understand this phenomenon in PRP therapy.

## 4. Materials and Methods

### 4.1. Preparation of PRP and Platelet Lysates

The study design and consent forms for all procedures (project identification code: 2019-0423) were approved by the Ethics Committee for Human Participants at Niigata University (Niigata, Japan) and complied with the Helsinki Declaration of 1964, as revised in 2013. Informed consent was obtained from all subjects involved in the study.

Blood samples were collected from non-smoking healthy male volunteers (*n* = 12, age: 28–63 years). Peripheral blood was collected in glass vacuum blood collection tubes (Vacutainer^®^; BD Biosciences, Franklin Lakes, NJ, USA) containing 1.5 mL A-formulation of acid–citrate–dextrose solution (ACD-A) (Terumo, Tokyo, Japan). Donors positive for HIV, HBV, HCV, or syphilis antibodies were excluded. After centrifugation, blood samples with optically recognized hemolysis or chylous plasma were also excluded.

As described previously [39], pure-platelet-rich plasma (P-PRP) was prepared by soft spin using horizontal centrifugation (415× *g*, 10 min). The upper plasma fraction (P-PRP) was further centrifuged by hard spin using fixed-rotor centrifugation (664× *g*, 3 min). Platelet pellets (per blood collection tube) were suspended in approximately 0.6 mL of the culture medium to add living platelets. Platelet lysates were prepared by 30 s sonication of platelets suspended in PBS, followed by top-speed centrifugation to exclude visible debris and filtration using sterile syringe filters (0.45 mm filter unit: Millex, Merck KGAA, Darmstadt, Germany).

### 4.2. Rat Dental Pulp-Derived Fibroblasts and Ultraviolet Irradiation

Rat dental pulp-derived line cells (RPC-C2A) were maintained in Dulbecco’s Modified Eagle Medium (DMEM) supplemented with 5% fetal bovine serum (FBS) (Gibco, Thermo Fisher, Waltham, MA, USA) at 37 °C in a CO_2_ incubator [48]. For ultraviolet irradiation, RPC-C2A cells were seeded at a density of 2.7–3.1 × 10^5^ cells/mL/well and precultured for 24 h. The medium was then replaced with the fresh medium containing 10 units/mL heparin sodium (Mochida Pharmaceutical Co., Ltd., Tokyo, Japan) to prevent fibrin clot formation [28] and irradiated by ultraviolet C (UVC) in a UVC chamber (UVST-1, Stanley Electric Co., Ltd., Tokyo, Japan) for 30 s. The UV radiation intensity was calculated to be 40.8 mJ/cm^2^ [47].

To UVC-irradiated RPC-C2A cells, human platelet suspensions in the medium (50 μL) were added to give a final concentration of 18–23 × 10^6^ platelets/mL/well. Platelet lysates in the medium prepared from the equivalent volume of platelet suspensions were added to the UVC-irradiated cells.

The number of RPC-C2A cells was determined by an automated cell counter using the Coulter principle (Moxi-Z; ORFLO, El Cajon, CA, USA). The basic protocols for cell treatment are summarized in Figure 9A.

### 4.3. Time-Lapse Videography

For time-lapse recording, RPC-C2A cells were seeded in a 40 mm dish and pretreated with 50 pg/mL demecolcine (FUJIFILM Wako Pure Chemical CO., Osaka, Japan) for 3 h to induce cell growth arrest [49], which was expected to avoid possible cell-division-dependent reduction in transferred mitochondria (per fibroblast) and to maintain extracellular spaces for easy follow-up of platelets. One hour after adding resting platelet suspensions in the absence or presence of ADP (3 μM), live imaging was initiated and recorded every 15 min for 24 h using a live-cell imager (CytoSMART Lux3; CytoSMART Technologies, Eindhoven, The Netherlands) in a CO_2_ incubator. Although ADP at this concentration is known to be a potent activator of platelets [50], this effectiveness was also confirmed in this study using immunofluorescence staining for CD62P expression and the platelet aggregation assay (Appendix A). Time-lapse video was constructed from serial images using image software (Adobe Premiere Elements 2022 (version 20.0), Adobe, San Jose, CA, USA).

Based on preliminary observations, we verified tiny black dots as platelets in this assay (Appendix A). As shown in Appendix A, no tiny dots were observed in the culture of the RPC-C2A cells alone (solo-culture). In contrast, such dots, which were almost identical to those observed in the culture of platelets alone, could be recognized in co-cultures, regardless of platelet status (Appendix A).

### 4.4. Visualization of Mitochondria Membrane Potential (Em)

RPC-C2A cells were pretreated with 0.1 mM carbonyl cyanide 4-(trifluoromethoxy)phenylhydrazone (FCCP) (Abcam, Cambridge, UK) [51] or 50 pg/mL demecolcine for 3 h to reduce the mitochondrial membrane potential or to inhibit cell division. Next, cells were treated with platelet suspensions in the absence or presence of 3 μM ADP for 24 h. At the end of the treatments, cells were stained using the MT-1 MitoMP Detection Kit (Dojindo Molecular Technologies, Inc., Kumamoto, Japan) without fixation and examined using a fluorescence microscope (Eclipse 80i; Nikon, Tokyo, Japan) connected with a cooled CCD camera (VB-7000; Keyence, Fukui, Japan) [52].

### 4.5. Image Analyses of the Color Images Obtained from the Visualization of Em

To enhance the levels of fluorescence signals for better recognition, the original color images were separated into RGB channels using image analysis software (WinRoof 2021 (version 5.3.0), Mitani Corp., Fukui, Japan). The resulting red images were converted to grayscale, and the background was reduced. Next, the brightness of representative single cells chosen randomly was quantified using WinRoof software.

### 4.6. Immunocytochemical Visualizations of Human Mitochondria Distribution

At the end of the treatment, RPC-C2A cells were fixed with a 10% formalin-neutral buffer solution, blocked with diluted Block Ace (1:2) (DS Pharma, Osaka, Japan), and treated with anti-human mitochondria antibody (1:100 dilution; Sigma-Aldrich, St. Louis, MO, USA) overnight at 4 °C, followed by probing with Alex Flour 555-conjugated anti-mouse IgG (1:200 dilution; BioLegend, San Diego, CA, USA). Nuclei were stained with DAPI (FUJIFILM Wako Pure Chemical CO.). After mounting with an antifade mounting medium (Vectashield; Vector Laboratories, Burlingame, CA, USA), the specimens were examined under a fluorescence microscope (ECLIPSE 80i; Nikon).

### 4.7. Visualization of Mitochondria Distribution Using MitoTracker

According to the protocol shown in Figure 9B, after 24 h of culture, RPC-C2A cells were carefully washed with a serum-free medium and labeled with 0.5 μM MitoTracker Green FM (ThermoFisher, Waltham, MA, USA) for 15 min in a CO_2_ incubator. After washing with the serum-containing medium, the cells were returned to the serum-containing medium. In parallel, platelets were washed with serum-free medium and labeled with 0.5 μM MitoTracker Orange CMTMRos (ThermoFisher) for 15 min at ambient temperature. After washing with the serum-free medium, platelets were resuspended in the serum-containing medium and added to the fibroblast cultures, as described previously. Between individual steps, platelets were centrifuged at 664× *g* for 3 min. At the end of the co-culture, the cells were fixed with 10% neutralized formalin for 5 min and examined using a fluorescence microscope with a fixed *z*-axis on the microscopic stage.

For preparation of UVC-irradiated platelets, platelets were irradiated for 60 s in the form of pure PRP and stained with MitoTracker Orange CMTMRos in the form of PBS suspension. After washing, platelets suspended in the culture medium were added to RPC-C2A cell cultures, as described above.

To obtain Z-stack images, RPC-C2A cells were co-cultured with MitoTracker-stained platelets for 24 h. The fibroblasts were detached from the plastic dish surface (and also weakly attached platelets) with 0.05% Trypsin plus 0.53 mM EDTA (FUJIFILM Wako Pure Chemical CO.), immobilized on glass slides using cytospin (Cytospin 4; Thermo Fisher Scientific, Waltham, MA, USA), and fixed with neutralized formalin. After membrane permeabilization with 0.1% Triton X-100, the fibroblasts were stained with DAPI. The specimens were examined under a fluorescence microscope after mounting with an antifade mounting medium (Vectashield; Vector Laboratories, Burlingame, CA, USA).

### 4.8. Scanning Electron Microscopy (SEM)

At the end of treatment, RPC-C2A cells were fixed with 2.5% neutralized glutaraldehyde, dehydrated, and freeze-dried, as described previously [53]. The rim of the dish was removed and examined by SEM using a TM-1000 microscope (Hitachi, Tokyo, Japan) at an accelerating voltage of 15 kV.

### 4.9. Determination of Fibroblast-Associated ATP Levels

At the end of the treatment, RPC-C2A cells were enzymatically detached, counted using an automated cell counter, and stored at −80 °C for several days before determining ATP levels. Cellular ATP levels were determined with a luminescence ATP assay kit (Dojin, Kumamoto, Japan) using a luminescencer (AB-2200, Atto Corp., Tokyo, Japan) [39]. Data were normalized to RPC-C2A cell counts.

### 4.10. Statistical Analysis

For multi-comparisons, one-way repeated-measures ANOVA was performed, followed by a Bonferroni test, using SigmaPlot 14.5 (Systat Software, Inc., San Jose, CA, USA). The dot plot in Figure 1 was created using KaleidaGraph 5.01 (Synergy Software, Reading, PA, USA). When the equal variance test (Brown–Forsythe) failed, one-way ANOVA on ranks was conducted, followed by the Tukey test, as shown in Figure 4H. This box plot was created using SigmaPlot 14.5.

To compare the two groups, the Wilcoxon signed-rank test was performed. A P-value of less than 0.05 was considered statistically significant. Data are presented as box plots drawn using SigmaPlot 14.5. Horizontal bars represent the median of the data.

## 5. Conclusions

Human platelets in the resting state can provide mitochondria to damaged rat dental pulp-derived fibroblasts, aiding their survival, likely through direct contact in a simple 2D co-culture system. This implies that mitochondrial transfer can revitalize damaged cells in cooperation with growth factors during tissue regeneration. More importantly, for PRP therapy, this simple co-culture system would be helpful as a preoperative test of the platelet potentials of individual PRP preparations, leading to improved current PRP therapies in the near future.

## Figures and Tables

**Figure 1 ijms-26-05504-f001:**
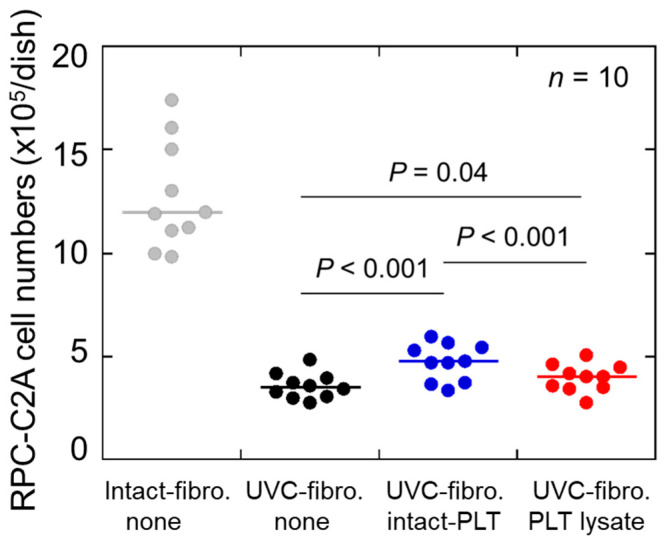
Rescuing effects of living platelets (PLTs) on UVC-irradiated RPC-C2A cells. After 24 h solo-culture and medium exchange, RPC-C2A cells were irradiated with UVC and cultured with platelets for 24 h. At the end of the treatment, cells were gently washed and counted for analysis of cell viability. Cell numbers without UVC irradiation were inserted in a blue square with a blue error bar. *n* = 10.

**Figure 2 ijms-26-05504-f002:**
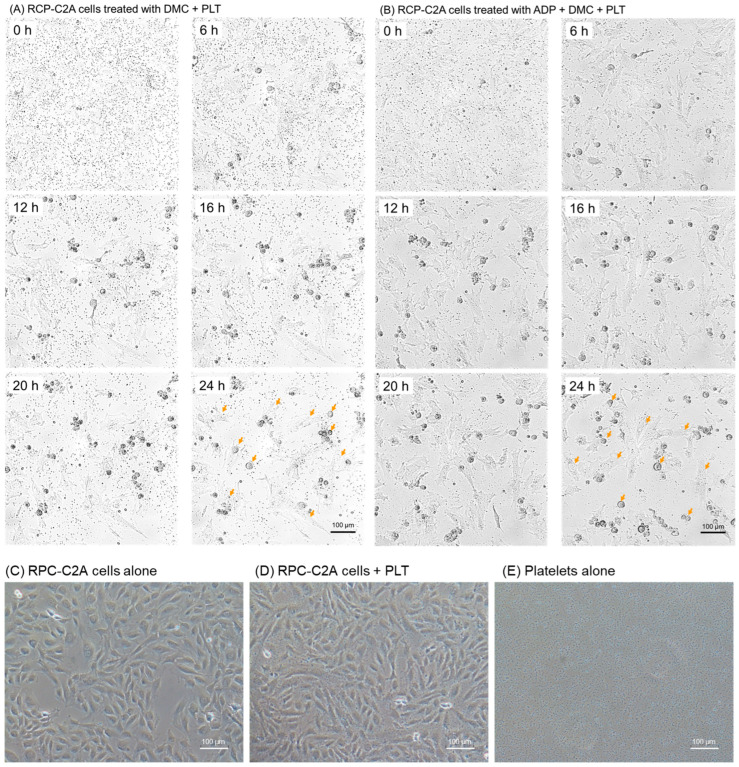
Time-course changes in platelet counts (observed as tiny black dots in extracellular spaces) in co-culture with RPC-C2A cells (orange arrows indicate some samples). After pretreatment with demecolcine (DMC), the cells were treated with living platelets in the absence (**A**) or presence (**B**) of ADP. Similar results were obtained from three independent experiments. Orange arrows indicate representative RCP-C2A cells at various growth phases. (**C**) RPC-C2A cells alone at 20 h. (**D**) RPC-C2A cells + platelets at 20 h. (**E**) Platelets alone at 20 h. Scale bar = 100 μm.

**Figure 3 ijms-26-05504-f003:**
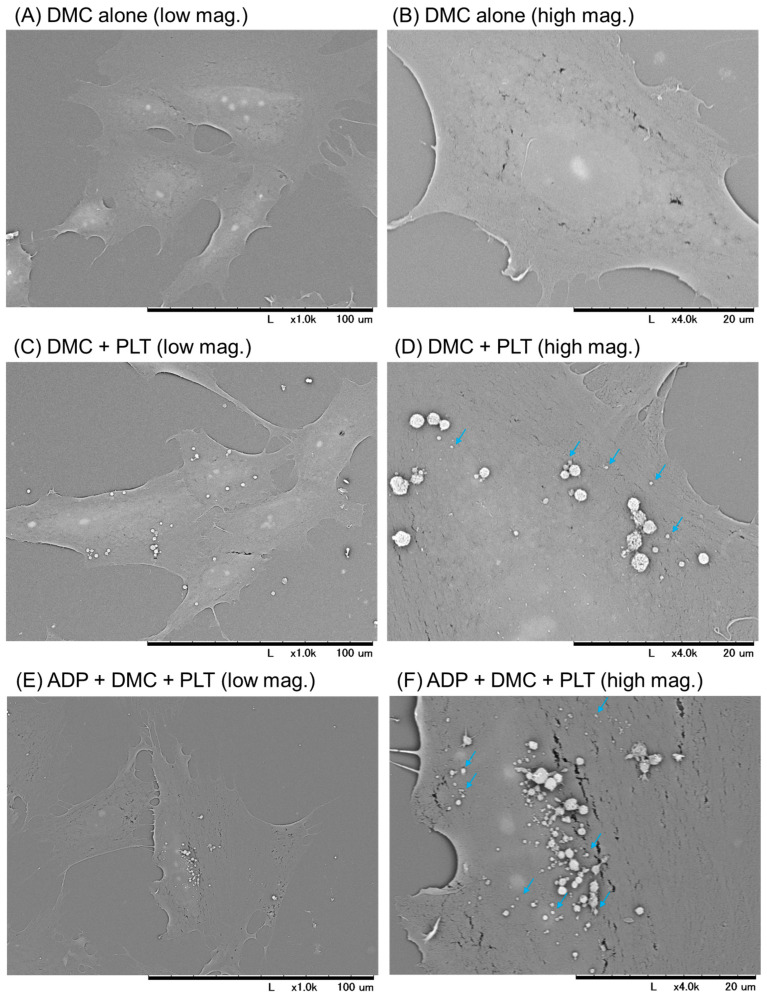
Photomicrographs by scanning electron microscopy of platelets and their microparticles on the surfaces of RPC-C2A cells. After pretreatment with demecolcine, cells were treated with living platelets (PLTs) for 24 h in the absence (**C**,**D**) or presence (**E**,**F**) of ADP. Control cells were pretreated with demecolcine (DMC) in the control (**A**,**B**). Similar findings were obtained from the other three independent experiments. Based on the difference in number, light blue arrows indicate representative microvesicles, probably released from activated platelets. Scale bar = 100 μm (low magnification) or 20 μm (high magnification).

**Figure 4 ijms-26-05504-f004:**
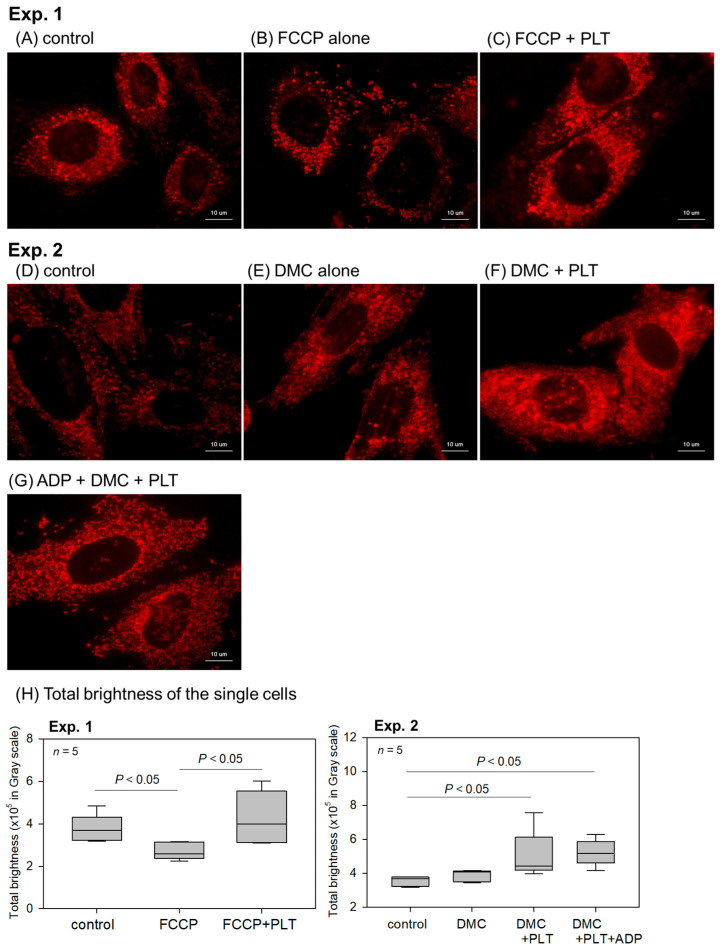
Effects of living platelets (PLTs) on the number of intact mitochondria indicated by membrane potential staining in RPC-C2A cells. In Experiment 1, after pretreatment with FCCP, cells were treated with living PLTs for 24 h in the absence (**B**,**C**) of ADP. In control (**A**), cells were not treated with inhibitors or PLTs. In Experiment 2, after pretreatment with demecolcine (DMC), cells were treated with living PLTs for 24 h in the absence (**E**,**F**) or presence of ADP (**G**). In control (**D**), cells were not treated with PLTs. Scale bar = 10 μm. (**H**) Quantification of the total brightness of each representative single cell. *n* = 5.

**Figure 5 ijms-26-05504-f005:**
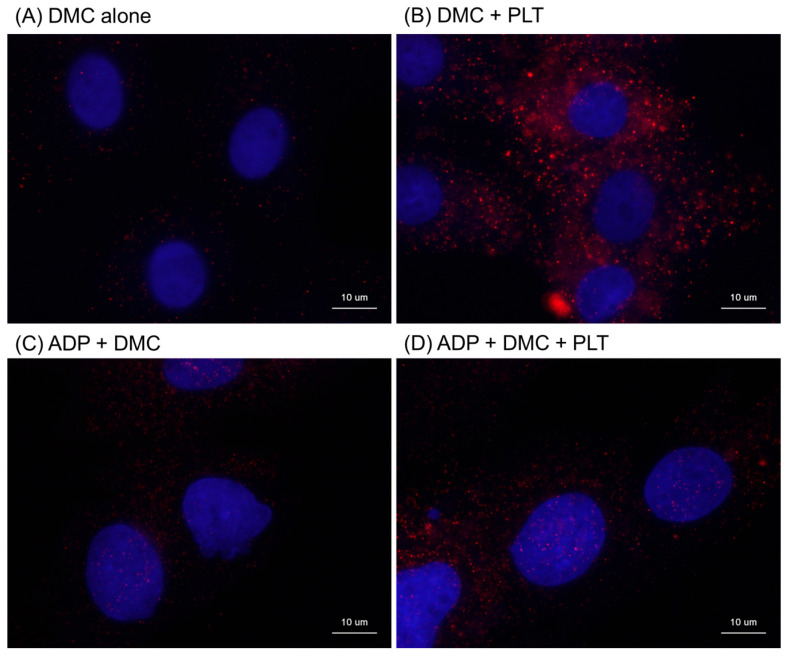
Effects of living platelets (PLTs) on the distribution of human platelet mitochondria in RPC-C2A cells. After pretreatment with demecolcine (DMC), the cells were treated with living PLTs for 24 h in the absence (**B**) or presence (**C**,**D**) of ADP. In the control (**A**), cells were pretreated with demecolcine alone. Human mitochondria and fibroblast nuclei were stained with anti-human mitochondria antibody, followed by visualization with secondary antibody (red), and DAPI (dark blue), respectively. Similar findings were obtained from the other three independent experiments. Scale bar = 10 μm.

**Figure 6 ijms-26-05504-f006:**
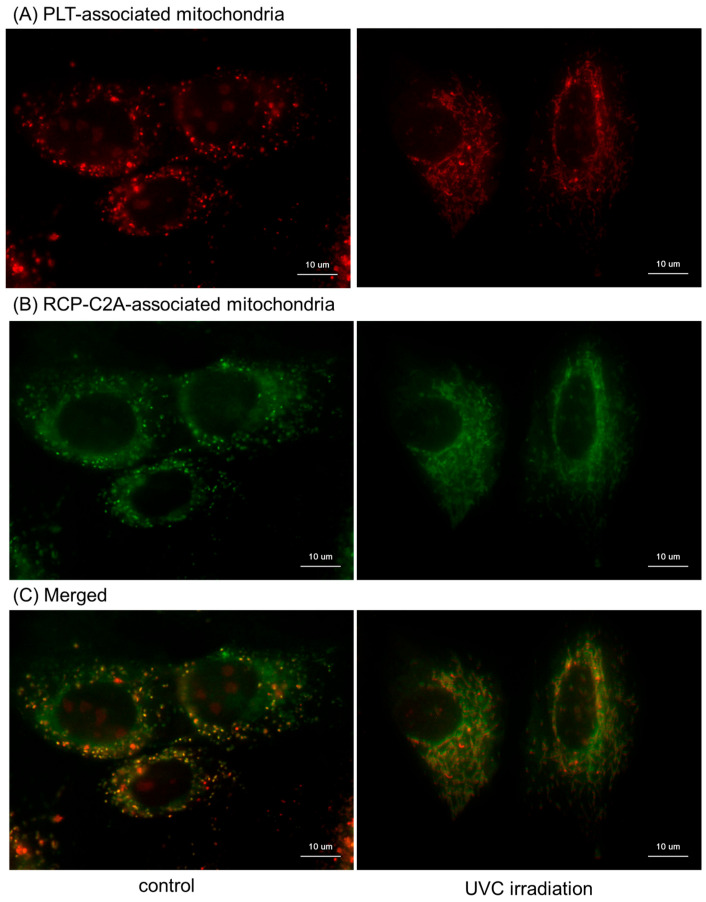
Distribution of platelet mitochondria transferred to RPC-C2A cells. Platelet and fibroblast mitochondria were labeled with MitoTracker Orange CMTMRos ((**A**): red) and MitoTracker Green FM ((**B**): green), respectively. After 24 h co-cultures, cells were gently washed, fixed with 10% neutralized formaldehyde, and examined. To observe individual distribution, both images were merged (**C**). Scale bar = 10 μm.

**Figure 7 ijms-26-05504-f007:**
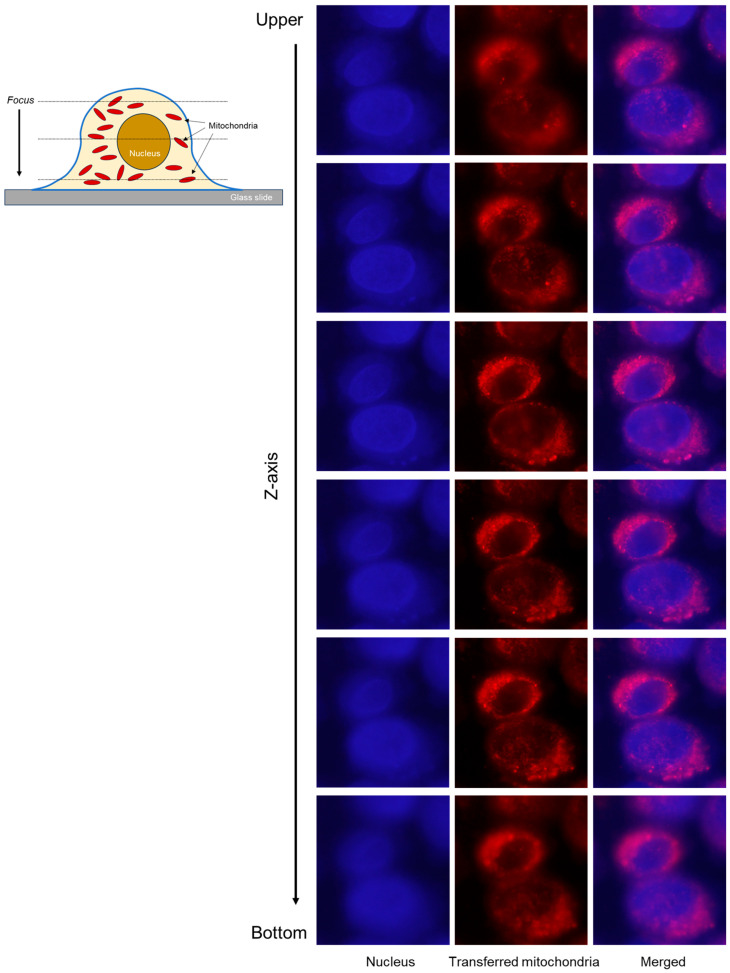
Z-stack images of RPC-C2A cells co-cultured with platelets, in which mitochondria were labeled with MitoTracker Orange CMTMRos (red). After 24 h of co-culture without other treatments, fibroblasts were enzymatically detached, pasted on glass slides using cytospin, fixed, and stained with DAPI (dark blue). As illustrated in the upper left panel, serial images were obtained using a fluorescence microscope along the *Z*-axis with an appropriate step size (approximately 0.15–0.2 μm) without changing the *X*- and *Y*-axes. Both images were merged to observe the mitochondrial distribution. Scale bar = 10 μm.

**Figure 8 ijms-26-05504-f008:**
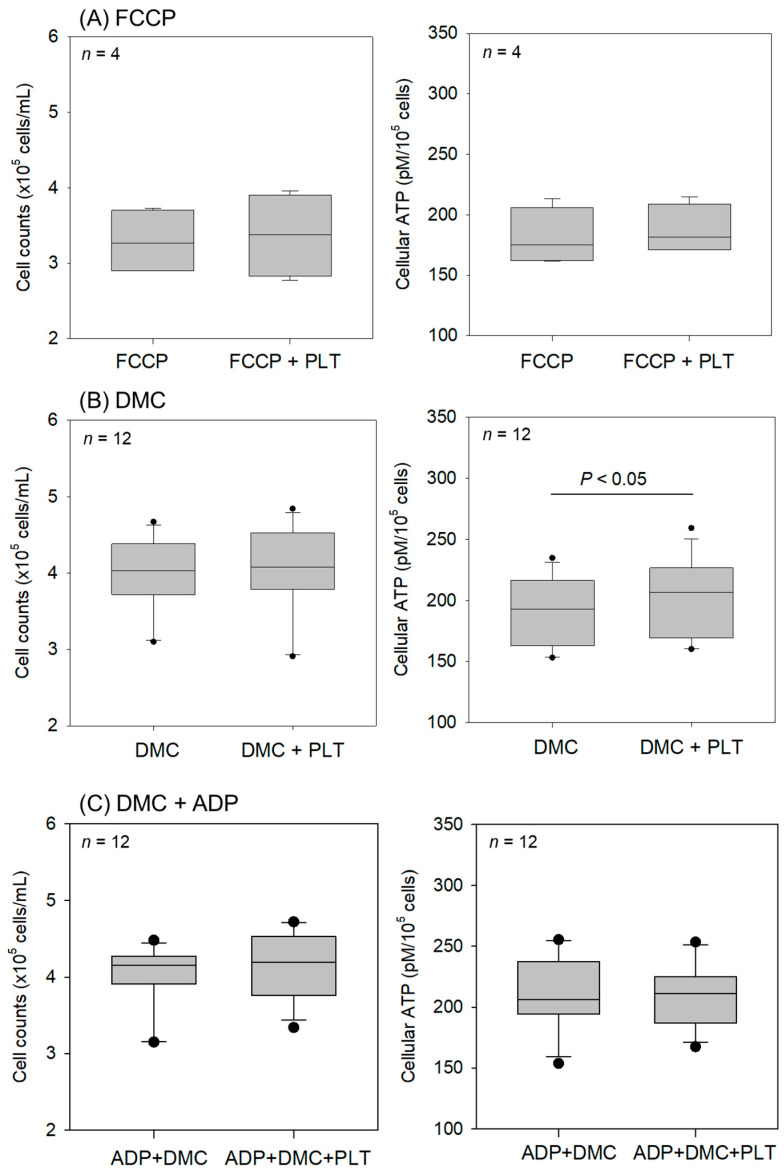
Effects of living platelets (PLTs) on intact RPC-C2A cell counts and cellular ATP levels. After pretreatment with FCCP (**A**) or demecolcine (DMC) (**B**,**C**), the cells were treated with living PLTs for 24 h in the absence (**A**,**B**) or presence (**C**) of ADP. *n* = 4 (**A**) and 12 (**B**,**C**).

**Figure 9 ijms-26-05504-f009:**
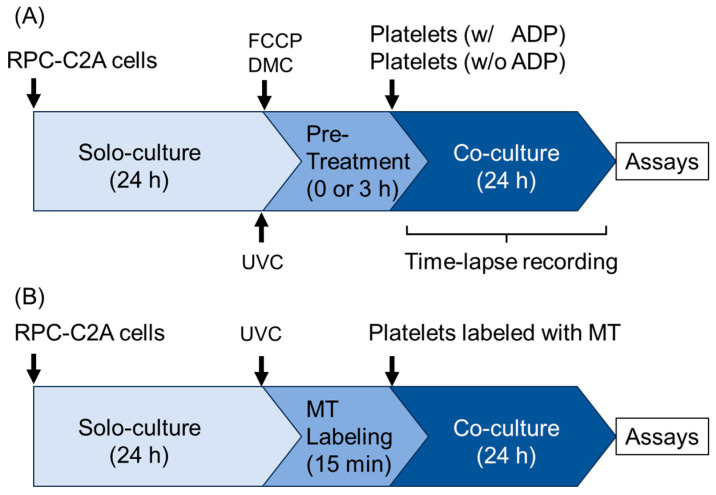
Experimental protocols for cell treatments. (**A**) Protocols for cell survival assay, time-lapse recording, mitochondrial membrane potential staining, immunofluorescence staining, and SEM examination. (**B**) Protocol for MitoTracker labeling assay. FCCP: carbonyl cyanide 4-(trifluoromethoxy)phenylhydrazone, DMC: demecolcine, UVC: ultraviolet-C, MT: MitoTracker. “Solo-culture” represents the culture of RPC-C2A cells alone without platelets.

## Data Availability

Data are available from the corresponding author upon request.

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
