# Peer review of "Mitochondrial Transfer from Human Platelets to Rat Dental Pulp-Derived Fibroblasts in the 2D In Vitro System: Additional Implication in PRP Therapy"

_ijms, 2025, doi:10.3390/ijms26125504_

Round 1
Reviewer 1 Report
Comments and Suggestions for Authors
The authors investigated the intercellular transfer of human platelet mitochondria to rat dental pulp-derived fibroblasts using a simplified two-dimensional co-culture system, with the goal of exploring the potential contribution of platelet mitochondria to tissue regeneration in platelet-rich plasma (PRP) therapy. The study is well-motivated and highlights a novel perspective on platelet function. However, several important limitations should be addressed to enhance the validity and depth of the findings:
- The study only included platelets from 12 healthy, non-smoking adult males. This narrow selection reduces the generalizability of the results. Future studies should incorporate a more diverse cohort to account for biological variability.
- The assessment of mitochondrial transfer relied primarily on mitochondrial membrane potential staining and immunofluorescence. Additional methods could provide more definitive evidence of active mitochondrial transfer.
- The co-culture duration was limited to 24 hours. This may be insufficient to capture to 24 hours.
- How the complex physiological factors may influence mitochondrial transfer, such as inflammatory cytokines or the three-dimensional architecture of tissues?
- Cross-species mitochondrial transfer could potentially elicit immune responses or disrupt cellular homeostasis, which may be discussed.
Comments on the Quality of English Language
The English could be improved to more clearly express the research.
Author Response
The authors investigated the intercellular transfer of human platelet mitochondria to rat dental pulp-derived fibroblasts using a simplified two-dimensional co-culture system, with the goal of exploring the potential contribution of platelet mitochondria to tissue regeneration in platelet-rich plasma (PRP) therapy. The study is well-motivated and highlights a novel perspective on platelet function. However, several important limitations should be addressed to enhance the validity and depth of the findings:
- The study only included platelets from 12 healthy, non-smoking adult males. This narrow selection reduces the generalizability of the results. Future studies should incorporate a more diverse cohort to account for biological variability.
Response: Thank you for this comment. We agree with you and think that the small sample size is a limitation of this study. However, at the same time, we expect that the capacity of mitochondrial transfer or platelet mitochondrial quality could be a promising criterion for predicting the prognosis of PRP therapy and help exclude non-responders in the future. We intend to continue this project until we have more definitive conclusions.
- The assessment of mitochondrial transfer relied primarily on mitochondrial membrane potential staining and immunofluorescence. Additional methods could provide more definitive evidence of active mitochondrial transfer.
Response: Thank you for this comment. We agree with you and hope to prove this possibility by using functional assays. In addition, if we can use a confocal microscope, the resulting data will be more convincing. However, we currently do not possess or use this powerful device. Thus, we manually operated an existing fluorescence microscope and attempted to obtain Z-stack images by modulating the microscope stage. Please see the revised Figure 7. Due to the enzymatic treatment for cell harvest, typical platelets with intense mitochondrial staining were not observed on the fibroblast surface area. Most platelet mitochondria appeared to be distributed in the cytoplasm adjacent to the nucleus, while some mitochondria overlapped the nucleus when the focus was observed above the nucleus.
- The co-culture duration was limited to 24 hours. This may be insufficient to capture to 24 hours.
Response: In a preliminary study using time-lapse recording with demecolcine, we confirmed that the extension of incubation time (~48 h) did not substantially change the fate of platelets, which continuously decreased in the co-cultures.
Regarding the cell survival test, the fibroblasts reached early confluency within 24 h without demecolcine (doubling time = approximately 18 h or less in DMEM+10% FBS). Thus, we concluded that this incubation time is not too short for this purpose.
In preliminary MitoTracker experiments, increasing the incubation time did not substantially increase MitoTracker staining in fibroblasts. Therefore, we fixed the incubation time at 24 h in this study. However, we hope to investigate further the time course and dose dependency in more detail.
- How the complex physiological factors may influence mitochondrial transfer, such as inflammatory cytokines or the three-dimensional architecture of tissues?
Response: Thank you for this comment. It has often been reported that mitochondrial transfer by itself contributes to inflammation resolution and tissue repair by enhancing the function of immune cells involved in regeneration, promoting cellular recovery, and restoring metabolic balance in damaged tissues [Arutusa et al., Immunol Lett 2025; 274:106992]. However, little information is available on the influence of inflammation on mitochondrial transfer. In cases of contact transfer and TNT-dependent transfer, inflammation may not severely interfere with the transfer. Conversely, inflammation could substantially influence this transfer when transferred via extracellular vesicles or in the naked form. This possibility should be investigated in future studies.
When and after our simple co-culture system is approved, published, and further improved, we intend to scale it up to a 3D co-culture system. As several probes may not be suitable for analyzing this complex system, we are now considering how to address this issue under our limited study conditions.
- Cross-species mitochondrial transfer could potentially elicit immune responses or disrupt cellular homeostasis, which may be discussed.
Response: Thank you for this comment. I was also anxious about immune rejection before. However, somehow, we have never experienced severe immune responses in animal studies using human PRP, which was pure PRP that excluded white blood cells (WBCs). To our knowledge, however, similar findings have not been published to date. Therefore, we would like to decline the expansion of the discussion until we reach any conclusions regarding this immunotolerance-like phenomenon in the near future. I am afraid that such an expansion may lead to readers’ misunderstanding or confusion. We hope to record our findings regarding mitochondrial transfer between cells of different animal species in a publication and hand over the baton for the subsequent study.
Reviewer 2 Report
Comments and Suggestions for Authors
PRP therapy for tissue regeneration and wound healing is an expanding field with many uncertainties due to heterogeneity of preparations, biological variability and frequent lack of standardization. In this study Nishiyama and coworkers propose that mitochondrial transfer contributes to the tissue regeneration properties of PRP and present a 2D cell culture model to address that claim. While the idea is intriguing, my main critique is that the authors do not present convincing experimental evidence of mitochondrial transfer taking place or platelets improving cell damage through their mitochondria. In view of that, I cannot recommend publication of this study.
Major
-It is exaggerated to describe the effects of platelets on irradiated fibroblasts (fig. 1) as a “rescue”. The effect is, at most, very modest. On lines 108-112, if normality and homoscedasticity are met, go ahead with a parametric test, there is no need to mention the result using a non-parametric test (they are usually more stringent). So to start with, we have here a really small effect of platelets and we cannot say yet whether this is due to their mitochondria.
-In the next experiment, on Figure 2, there are clearly changes in the numbers of platelets (tiny dots). In addition, the fibroblasts appear to change morphology and become round; does this represent cells undergoing apoptosis? Controls with only fibroblasts and only platelets in the same conditions are missing.
-Figure 3 shows platelets (and microvericles) attached to the surface of fibroblasts (only to 10% of them, authors claim). Obviously, this does not prove mitochondrial transfer.
-Figure 4 does not determine numbers of mitochondria. Any quantification (based in fluorescence intensity) is presented in Figure 5. In fact, both figures should be combined into one and there is no need to present color (red) and grayscale panels, the color is irrelevant and on top of that red tends to be difficult to see.
This said, these experiments lack critical controls in which platelet mitochondria have been inactivated, to determine whether any increase in intensity is due to the mitochondria provided by the platelets.
Figure 5C, this needs proper quantification of sufficient individual cells across the three independent experiments the authors claim to have done, plus statistical analysis.
-In the next experiment (fig. 6), in the legend, indicate how the human mitochondria were visualized (antibody staining). I assume DAPI was used to visualize nuclei, that should also be mentioned. Along with figure 7, what evidence is this of platelet mitochondria being inside the fibroblasts? The authors seem to have use an Eclipse 80i microscope. Did they perform confocal microscopy, which I don’t get the impression they did? Detail is missing. State of the art microscopy (super resolution) would be critical to demonstrate the hypothesized mitochondrial transfer. As it stands, the images probably represent mitochondria of platelets attached to the cell surface (as shown in the electron micrographs of fig. 3).
-The data of figure 8 can also be explained by mitochondria of the platelets attached to the fibroblast cell surface.
-The discussion has contradicting statements: mitochondria ere transferred (lines 240-241) and no direct evidence of transfer (line 248). Decrease of platelet counts in co-cultures is still no proof; they can have lysed during the experiment.
Minor
-The introduction needs more references. For example, lines 54-58; lines 63-65.
-Line 45-46, I don’t understand this sentence
-Line 79, what is it meant with “it is physically challenging to speculate…”
-Lines 87-90, it would certainly be a paradigm shift, but not just for PRP therapy.
-Line 101-102, expanding knowledge about mitochondria transfer is not clear justification for using rat dental pulp fibroblasts. Maybe you have an interest in odontology applications of PRP and that would be a better justification.
-Line 126, for a total of 24 h.
-Lines 190-191, more accurate: rat cells did not display staining for human mitochondria, or something similar.
-Line 197: distribution of human platelet mitochondria
-Line 309, chylous plasma
-Line 316, specify how lysates were filtered
-Lines 321 and 328, clarify number of cells added per well. Was it a full ml and therefore 2.7-3.1 x 10E5 cells in total?
Comments on the Quality of English Language
The language is at good level, a few places would require attention but probably this could be ressolved at the tiume of typesetting, provided the paper is accepted
Author Response
PRP therapy for tissue regeneration and wound healing is an expanding field with many uncertainties due to heterogeneity of preparations, biological variability and frequent lack of standardization. In this study Nishiyama and coworkers propose that mitochondrial transfer contributes to the tissue regeneration properties of PRP and present a 2D cell culture model to address that claim. While the idea is intriguing, my main critique is that the authors do not present convincing experimental evidence of mitochondrial transfer taking place or platelets improving cell damage through their mitochondria. In view of that, I cannot recommend publication of this study.
Response: Thank you for your consideration. Even if you meant that data obtained using a confocal microscope is “must-present,” we do not possess this device. We could not use a common facility within 10 days, which was granted for revision. Thus, we cannot conduct such a sophisticated experiment for this revision.
All the data we can present in the original version may have seemed “circumstantial” or “phenomenological” to you. Although a confocal microscope was unavailable, we have conducted additional experiments using a conventional microscope to make our best responses and revisions in this revision to address your concern. We hope that you appreciate Figure 7.
Major
- It is exaggerated to describe the effects of platelets on irradiated fibroblasts (fig. 1) as a “rescue”. The effect is, at most, very modest. On lines 108-112, if normality and homoscedasticity are met, go ahead with a parametric test, there is no need to mention the result using a non-parametric test (they are usually more stringent). So to start with, we have here a really small effect of platelets and we cannot say yet whether this is due to their mitochondria.
Response: Based on your comment, we have deleted a note from the non-parametric analysis. Additionally, to address your critique, we have included the significance of these data in the Results section: these data indicate not mitochondrial transfer, but rather the existence of unidentified factors beyond platelet growth factors that contribute to tissue regeneration.
We understand your concern about the rescue effects. Therefore, we have described the data presented here as indirect and phenomenological evidence, rather than direct and convincing evidence. However, we hope that the sum of the following data supports this possibility and at least does not exclude the consistency. Please see also our response to your last major comment.
We have added an expression of “modest” to the Result section regarding the levels of the recurring effects.
- In the next experiment, on Figure 2, there are clearly changes in the numbers of platelets (tiny dots). In addition, the fibroblasts appear to change morphology and become round; does this represent cells undergoing apoptosis? Controls with only fibroblasts and only platelets in the same conditions are missing.
Response: The round-shaped fibroblasts you indicated were a result of treatment with demecolcine, which was expected to synchronize the cell cycle and suppress cell division, because demecolcine depolymerizes microtubules [Saraiva et al., Cloning Stem Cells 2009; 11(1):141-52]. To avoid a possible cell-division-dependent reduction of transferred mitochondria (per fibroblast) and maintain the extracellular spaces for clear recording of tiny dots, i.e., platelets, we considered using this type of reagent. We have added this explanation in the Methods and Results sections.
In this revision, to address your question, we have also added the photos of fibroblasts in the absence of demecolcine for comparison. However, due to the limited time granted for revision, we were unable to conduct the apoptosis assay this time.
- Figure 3 shows platelets (and microvericles) attached to the surface of fibroblasts (only to 10% of them, authors claim). Obviously, this does not prove mitochondrial transfer.
Response: As you claimed, this data is not meant to prove mitochondrial transfer. This SEM examination aimed to explore the mechanism(s) of mitochondrial transfer. According to the widely accepted theory, microvesicles released from platelets are thought to primarily transport mitochondria to the surrounding cells. This data shows microvesicles existed around activated platelets on the fibroblast cell surface. However, the other data in this study did not support the possibility that activated platelets increase the platelet-derived mitochondria-staining, i.e., mitochondrial transfer, in the fibroblasts.
- Figure 4 does not determine numbers of mitochondria. Any quantification (based in fluorescence intensity) is presented in Figure 5. In fact, both figures should be combined into one and there is no need to present color (red) and grayscale panels, the color is irrelevant and on top of that red tends to be difficult to see. This said, these experiments lack critical controls in which platelet mitochondria have been inactivated, to determine whether any increase in intensity is due to the mitochondria provided by the platelets.
Response: We have combined Figures 4 and 5 and removed several unnecessary images, including grayscale ones, to explain data processing.
To address your comment that these experiments lacked critical controls, we further presented additional data obtained from experiments using different probes to visualize platelet mitochondria. These data did not stand alone but should be evaluated with the overall dataset.
- Figure 5C, this needs proper quantification of sufficient individual cells across the three independent experiments the authors claim to have done, plus statistical analysis.
Response: Based on your suggestions, we have further analyzed the images, quantified their fluorescence intensity, and modified the graph to include statistical analysis. Although unexpected changes in the background levels may have affected the pixel values, the relative levels among the groups in Figure 4 seem consistent with those of the previous data (Fig. 5).
- In the next experiment (fig. 6), in the legend, indicate how the human mitochondria were visualized (antibody staining). I assume DAPI was used to visualize nuclei, that should also be mentioned. Along with figure 7, what evidence is this of platelet mitochondria being inside the fibroblasts? The authors seem to have use an Eclipse 80i microscope. Did they perform confocal microscopy, which I don’t get the impression they did? Detail is missing. State of the art microscopy (super resolution) would be critical to demonstrate the hypothesized mitochondrial transfer. As it stands, the images probably represent mitochondria of platelets attached to the cell surface (as shown in the electron micrographs of fig. 3).
Response: Based on your suggestion, we have added notes on immunological staining of human mitochondria and nuclear staining using DAPI in the legend of Figure 6.
We attempted to demonstrate 3D mitochondrial distribution in single fibroblasts with our limited devices. There is no doubt that the confocal microscope is a powerful tool for observing inside cells without destruction. However, we could not freely use such a device, and thus we showed the presence of platelets attached to the fibroblast surface using scanning electron microscopy (SEM) in the original manuscript. As shown in Figure 3, some platelets appeared on the fibroblast surface; however, the number of such platelets was insufficient to account for the increases in mitochondrial staining. Thus, despite the “indirect” nature, we tentatively concluded that platelet mitochondria were incorporated into the fibroblast.
In the revised version, we have performed additional experiments using our existing fluorescence microscope to obtain Z-stack images. Please see Figure 7. Although we could not reconstruct the 3D images or completely exclude the platelets possibly existing on the fibroblast surface, we believe these data are sufficient to support mitochondrial transfer.
- The data of figure 8 can also be explained by mitochondria of the platelets attached to the fibroblast cell surface.
Response: This may indicate the possible involvement of the attached platelets in ATP levels. However, as described in the previous article [Melchinger et al., Frontiers in Cardiovasc Med 2019; 6:153], each platelet possesses 5-8 mitochondria, whereas fibroblasts seemingly possess thousands of mitochondria. In our experience, we have usually found that each platelet includes 3-6 mitochondria.
Thus, to influence cellular ATP levels and visualization, at least hundreds of platelets must be attached to the fibroblast cell surface. Nevertheless, as shown by the SEM observations, the number of attached platelets in our study did not exceed 10 (per fibroblast). Thus, we believe that attached platelets are less influential in this quantitative comparison.
- The discussion has contradicting statements: mitochondria ere transferred (lines 240-241) and no direct evidence of transfer (line 248). Decrease of platelet counts in co-cultures is still no proof; they can have lysed during the experiment.
Response: We aimed to note that the data we obtained here could be considered circumstantial. Unfortunately, we were unable to obtain more convincing data under our limited experimental conditions and devices.
Even though platelets were lysed within 24 h, we can consider this phenomenon as a “decrease in platelet number.” Judging from many published articles, platelets are activated during and after mitochondrial transfer, especially when extracellular vesicles (EVs) are released, and are generally in the process of dying. Thus, the number of platelets could be decreased by autolysis. Further investigation is required to address your critique and demonstrate the occurrence of autolysis. However, our description was not theoretically incorrect.
Minor
- The introduction needs more references. For example, lines 54-58; lines 63-65.
Response: We have added 15 references.
- Line 45-46, I don’t understand this sentence
Response: We aimed to highlight the trend surrounding the PRP boom during its early phase (about1995-2005). Clinicians who immediately used PRP for their patients believed the “sales pitch” that PRP acts predominantly as a growth factor cocktail at injured sites. Based on this comment, we have modified this sentence.
- Line 79, what is it meant with “it is physically challenging to speculate…”
Response: As described in the sentences preceding this one, platelets rely on glycolysis for energy generation, with the ratio of glycolysis to mitochondrial respiration being crucial. Thus, even though many platelets can be collected, it is questionable whether a sufficient number of mitochondria can be provided to the surrounding cells. This is what it meant. We have modified a little.
- Lines 87-90, it would certainly be a paradigm shift, but not just for PRP therapy.
Response: Based on this comment, we have deleted the expression that reminds of PRP therapy. Instead, we have added the possible modification of PRP preparation and application protocols.
- Line 101-102, expanding knowledge about mitochondria transfer is not clear justification for using rat dental pulp fibroblasts. Maybe you have an interest in odontology applications of PRP and that would be a better justification.
Response: The primary purposes of using rat dental pulp cells were to demonstrate mitochondrial transfer between different animal species and subsequently to show the potential emergency use of human PRP in pet animals. Currently, this may be considered entirely nonsensical; however, PRP may still be applied to humanized animals.
This approach is not limited to dental problems, but can also be applied to other tissue regeneration and wound healing, including bleeding.
However, as described in the text, the scientific purpose of this study was to establish a simple co-culture system suitable for investigating the transfer of platelet mitochondria.
- Line 126, for a total of 24 h.
Response: We have corrected it.
- Lines 190-191, more accurate: rat cells did not display staining for human mitochondria, or something similar.
Response: We have corrected it.
- Line 197: distribution of human platelet mitochondria
Response: We have corrected it.
- Line 309, chylous plasma
Response: We have corrected it.
- Line 316, specify how lysates were filtered
Response: We have added the information about the syringe filter we used.
- Lines 321 and 328, clarify number of cells added per well. Was it a full ml and therefore 2.7-3.1 x 10E5 cells in total?
Response: Yes, it was. We conducted co-culture in 40-mm dishes with 1 mL culture medium.
Round 2
Reviewer 1 Report
Comments and Suggestions for Authors
I have no additional comments.
Comments on the Quality of English Language
The English could be improved to more clearly express the research.
Reviewer 2 Report
Comments and Suggestions for Authors
The authors have made a great effort at addressing my critique points. I'm still not totally convinced about the results allowing a definitive answer to the question of platelet mitochondria transfer, so any conclusion in the paper must acknowledge the technical limitations, which I think the authors now do better. I hope future studies using more precise techniques will provide a definitive answer.